# The Impact of Tea Consumption on Prediabetes Regression and Progression: A Prospective Cohort Study

**DOI:** 10.3390/nu17142366

**Published:** 2025-07-19

**Authors:** Tingting Li, Christopher K. Rayner, Michael Horowitz, Karen Jones, Cong Xie, Weikun Huang, Zilin Sun, Shanhu Qiu, Tongzhi Wu

**Affiliations:** 1Department of Endocrinology, Zhongda Hospital, Institute of Diabetes, School of Medicine, Southeast University, Nanjing 210009, China; 230219007@seu.edu.cn (T.L.); 101007988@seu.edu.cn (Z.S.); 2Centre of Research Excellence in Translating Nutritional Science to Good Health, Adelaide Medical School, The University of Adelaide, Adelaide, SA 5000, Australia; chris.rayner@adelaide.edu.au (C.K.R.); michael.horowitz@adelaide.edu.au (M.H.); karen.jones@adelaide.edu.au (K.J.); c.xie@adelaide.edu.au (C.X.); weikun.huang@adelaide.edu.au (W.H.); 3Department of General Practice, Zhongda Hospital, Institute of Diabetes, School of Medicine, Southeast University, Nanjing 210009, China

**Keywords:** tea consumption, prediabetes regression, prediabetes progression

## Abstract

**Background:** Lifestyle modifications are pivotal to preventing the progression of prediabetes and associated cardiometabolic diseases. Recent evidence from cross-sectional analysis of community-dwelling Chinese adults suggests that regular consumption of tea, particularly dark tea, is associated with a reduced risk of both prediabetes and type 2 diabetes. However, the effects of tea consumption on prediabetes progression and regression remain uncertain. This study investigated the associations of tea consumption with prediabetes progression and regression in Chinese adults with prediabetes. **Methods:** A cohort of 2662 Chinese adults with prediabetes was followed over ~3 years. Baseline tea consumption, including the type (green, black, dark, or other) and frequency (daily, sometimes, or nil), was assessed using standardized questionnaires. Prediabetes was defined according to the American Diabetes Association criteria. Multinomial logistic and linear regression analyses with multivariable adjustment was performed to evaluate associations. **Results:** Compared to non-tea drinkers, dark tea consumers were less likely to progress to type 2 diabetes (odds ratio [OR]: 0.28, 95% confidence interval [CI]: 0.11, 0.72, *p* = 0.01), whereas green tea consumption was associated with a reduced probability of regressing to normoglycemia (OR: 0.73, 95 CI%: 0.59, 0.90, *p* = 0.01). **Conclusions:** These findings support further exploration of dark tea consumption as a strategy to reduce prediabetes progression, and suggest that effects of green tea consumption should also be examined more closely in this population.

## 1. Introduction

Prediabetes is a highly prevalent metabolic disorder worldwide, and is estimated to affect about one third of adults in the community in China [1,2]. Mounting evidence suggests that prediabetes represents a major independent risk factor for cardiovascular disease and mortality [3,4], and that this risk can be mitigated by regression to normoglycemia [5]. International diabetes guidelines emphasize lifestyle interventions to prevent prediabetes progression and promote its regression [6,7]. In relation to dietary factors, tea consumption is generally considered to exhibit a favorable impact on glucose homeostasis [8,9]. However, we and others have provided evidence that the type of tea consumed is important [9,10,11], probably reflecting differences in the tea manufacturing process and resultant biological effects [12].

In China, tea is commonly classified according to the specialized manufacturing processes, particularly the degree of enzymatic oxidation (often referred to as fermentation) [13]. Table 1 outlines the distinct differences in production methods and compositional characteristics among green, black, and dark teas. Green tea undergoes minimal processing, with rapid heat treatment (steaming or pan-firing) applied post-harvest to inactivate oxidation enzymes. This preserves its verdant color and high concentrations of catechins, particularly epigallocatechin gallate, which are potent antioxidants and are widely perceived to drive potential health benefits [14,15]. While some studies suggest that daily green tea consumption (more than 6 cups per day) is associated with a decreased risk of type 2 diabetes [16,17], others show inconsistent findings [18,19,20]. For example, a study in Vietnam reported that higher green tea intake was linked to increased risks of diabetes and insulin resistance [18]. Similarly, a prospective study in Chinese adults found that green tea drinkers had a 20% higher risk of diabetes compared to non-drinkers [19]. Relative to green tea, semi-fermented teas, such as Oolong tea, undergo partial enzymatic oxidation, resulting in more complex aroma and flavor due to the partial conversion of catechins into theaflavins and thearubigins [21]. Black tea, by contrast, undergoes complete enzymatic oxidation, thus extensively converting catechins into theaflavins and thearubigins [22]. A prospective cohort study among Japanese male workers found that consumption of two or more cups of Oolong tea per day was associated with an increased risk of diabetes [11]. In contrast, a prospective study in Singapore Chinese adults reported that drinking more than one cup of black tea per day was associated with a decreased risk of type 2 diabetes [10]. Dark tea differs markedly in its production, undergoing a unique post-fermentation process involving microbial activity [23,24]. This results in substantial alterations in the abundance and composition of bioactive compounds, including alkaloids, free amino acids, polyphenols, polysaccharides and their derivatives [24], which may underpin its favorable cardiometabolic effects. In our recent cross-sectional analysis of a Chinese adult population, we found that regular dark tea consumption was strongly associated with an increase in urinary glucose excretion and a reduction in the risk of prediabetes and type 2 diabetes [9].

While a number of cohort studies have reported inconsistent findings on the association of tea consumption with incident type 2 diabetes [17,25,26], there is a lack of information relating to the impact of tea consumption on prediabetes progression and regression. This prospective study investigated the associations of type and frequency of tea consumption with prediabetes progression and regression in a large cohort of Chinese community-dwelling adults with prediabetes.

## 2. Materials and Methods

### 2.1. Participants

This study represents a prospective analysis of a subset of data derived from the Study on Evaluation of iNnovative Screening tools and determInation of optimal diagnostic cut-off points for type 2 diabetes in Chinese multi-Ethnic (SENSIBLE) cohort, which enrolled a nationally representative sample of community-dwellers aged ≥ 18 years from 8 provinces to determine the optimal cut-off values for advanced glycation end-products and glycated hemoglobin (HbA1c) for the diagnosis of type 2 diabetes in China [27,28]. A total of 3715 participants with prediabetes were surveyed on the type and frequency of tea consumption using standardized questionnaires (detailed in the data collection section) at baseline between November 2016 to June 2017 and followed over ~3 years. After excluding participants who had provided insufficient information about the types of tea consumed (n = 97), habitual consumption of ≥2 types of tea (n = 200) and those lost to follow-up (n = 756), 2662 participants were included in the final analysis (Figure 1). The protocol of the SENSIBLE cohort study was approved by the Human Research Ethics Committee of Zhongda Hospital, Southeast University, Nanjing, China (approval number: 2016ZDSYLL092-P01; approval date: 11 January 2017). Written informed consent was obtained from all participants.

### 2.2. Data Collection

At baseline, information relating to tea consumption (including the type and frequency), dietary habits (including salt and fat content, vegetable and fruit consumption categorized as minimal, low, moderate, or high), regular exercise, antihypertensive medication, family history of diabetes, smoking, and alcohol consumption were collected using a standardized questionnaire by trained interviewers. Body weight, height, waist circumference, and systolic and diastolic blood pressure (SBP and DBP) were measured according to standardized protocols, as reported previously [9,29]. At both baseline and follow-up visits, fasting venous blood was collected after an overnight fast (>8 h) for measurements of fasting plasma glucose (FPG), HbA1c, total cholesterol, triglycerides (TG), high-density lipoprotein-cholesterol (HDL-C), low-density lipoprotein-cholesterol (LDL-C) and serum creatinine. Plasma glucose at 2 h after a 75 g oral glucose drink (2hPG) was also measured in participants without known diabetes. Insulin resistance was assessed by the triglyceride-and-glucose (TyG) index [30,31]. The estimated glomerular filtration rate (eGFR) was calculated using the Chronic Kidney Disease Epidemiology Collaboration formula for the Chinese population [32].

Based on the American Diabetes Association criteria [33], prediabetes was defined as FPG 5.6–6.9 mmol/L, 2hPG 7.8–11.0 mmol/L during the 75 g oral glucose tolerance test or HbA1c 39–47 mmol/mol (5.7–6.4%), while diabetes was defined as FPG ≥ 7.0 mmol/L, 2hPG ≥ 11.1 mmol/L, HbA1c ≥ 48 mmol/mol (6.5%), self-reported history or the use of glucose-lowering medication for established type 2 diabetes, and normoglycemia was defined as FPG < 5.6 mmol/L, 2hPG < 7.8 mmol/L and HbA1c < 39 mmol/mol (5.7%). The type of tea was classified as green, black, dark, or other. The frequency of tea consumption was categorized as (1) nil, (2) occasional (at least once a month), (3) frequent (at least once a week), or (4) daily. The numbers of participants who drank tea either occasionally or frequently proved to be limited, so we combined these two groups for analysis and defined the frequency as ‘sometimes’.

### 2.3. Statistical Analysis

Continuous variables are presented as means ± standard deviations and categorical variables as numbers (percentages). Participants were stratified into three groups over follow-up, who (i) progressed to diabetes, (ii) regressed to normoglycemia, or (iii) remained as prediabetes. Differences across the three groups at baseline were compared using one-way analysis of variance, or χ^2^ tests, as appropriate, with post hoc comparisons adjusted by Bonferroni’s correction. Multinominal logistic regression analysis was performed to evaluate the associations between tea consumption and prediabetes progression and regression in both the crude model (Model 1) and adjusted model (Model 2, adjusted for age, sex, BMI, mean arterial pressure, TC, HDL-C, LDL-C, eGFR, low-salt low-fat diet, regular exercise, antihypertensive medication, family history of type 2 diabetes, current smoking status, current alcohol consumption, vegetable consumption and fruit consumption). Linear regression analysis was used to assess the associations of insulin resistance with tea consumption. A variance inflation factor (VIF) > 10 was taken to indicate collinearity between variables [34], but no significant collinearity was detected in any of our analyses. Subgroup analyses based on gender and ethnicity differences were also performed to assess these associations. All statistical analyses were conducted using SPSS (version 25.0, IBM, New York, USA). *p* values < 0.05 were considered statistically significant.

## 3. Results

### 3.1. Baseline Characteristics

A total of 2662 participants with prediabetes (Table 2) were included in the final analysis. Of these, 1647 remained as prediabetes, 299 progressed to type 2 diabetes, and 716 regressed to normoglycemia during the 3-year follow-up. Compared with participants who remained as prediabetes, the group who progressed to diabetes had a slightly higher proportion of males, and higher WC, blood pressure, FPG, 2hPG, HbA1c, TG, use of hypotensive medication, and proportion of current smokers, but similar age, BMI, TC, HDL-C, LDL-C, eGFR, adherence to low-salt low-fat diet and regular exercise, family history of diabetes, habitual alcohol intake, vegetable consumption, and fruit consumption. By contract, participants who regressed to normoglycemia were younger and had lower BMI, WC, blood pressure, FPG, 2hPG, HbA1c, TC, TG, LDL-C, and less regular exercise, use of hypotensive medications, and habitual alcohol intake at baseline (all *p* < 0.05).

Appendix A presents baseline characteristics of participants stratified by tea type. Of the total cohort, 1198 were non-tea drinkers, while 1016 consumed green tea, 138 consumed black tea, 140 consumed dark tea, and 170 consumed other types of tea. Compared to non-tea drinkers, consumers of green, black and dark tea were generally younger and had a higher BMI, WC, and lower FPG. No significant differences were observed across the groups in BP, 2hPG, HbA1c, TC, LDL-C or HDL-C.

### 3.2. Tea Consumption and Prediabetes Progression and Regression

Associations of tea consumption with prediabetes regression and progression are shown in Table 3. When compared with non-tea drinkers, those who consumed dark tea, but not green, black or other tea, exhibited lower odds of progressing to diabetes (odds ratio [OR]: 0.30, 95% confidence interval [CI]: 0.12 to 0.75, *p* = 0.01) in the unadjusted model (Model 1). After multivariable adjustment (Model 2), the OR for dark tea drinkers was 0.28 (95% CI: 0.11 to 0.72, *p* = 0.01). By contract, participants who consumed green tea, but not other types, exhibited lower odds of regressing to normoglycemia (OR: 0.76, 95% CI: 0.63 to 0.93, *p* = 0.01) in the crude model (Model 1). After multivariable adjustment (Model 2), this association remained significant (OR: 0.73, 95% CI: 0.59 to 0.90, *p* = 0.01). Subgroup analyses revealed that the association between dark tea consumption and reduced odds of prediabetes progression was particularly evident in females (OR: 0.27, 95% CI: 0.08 to 0.90, *p* = 0.03) (Appendix A) and non-Han subgroups (OR: 0.18, 95% CI: 0.04 to 0.80, *p* = 0.02) (Appendix A), and that the association between green tea consumption and reduced odds of prediabetes regression was more evident in females (OR: 0.76, 95% CI: 0.58 to 0.99, *p* = 0.04) (Appendix A) and Han subgroups (OR: 0.72, 95% CI: 0.58 to 0.90, *p* = 0.01) (Appendix A).

### 3.3. Frequency of Tea Consumption and Prediabetes Progression and Regression

Extending the findings in Table 3, the associations between the frequency of consumption of both green and dark tea and prediabetes progression and regression were further examined, and are summarized in Table 4. Compared with non-tea drinkers, those who consumed dark tea daily exhibited lower odds of progressing to diabetes in both unadjusted and multivariable-adjusted models (model 1: OR: 0.22, 95% CI: 0.07 to 0.71, *p* = 0.01 and model 2: OR: 0.22, 95% CI: 0.07 to 0.71, *p* = 0.01, respectively). In participants who consumed green tea daily, the odds of regressing to normoglycemia were reduced (model 1: OR: 0.74, 95% CI: 0.59 to 0.94, *p* = 0.01 and model 2: OR: 0.72, 95% CI: 0.56 to 0.92, *p* = 0.01, respectively).

### 3.4. Tea Consumption and Insulin Resistance

As shown in Table 5, consumption of green tea was associated with an increase in TyG in both unadjusted and multivariable-adjusted models (0.08; 95% CI: 0.03 to 0.13, *p* = 0.001 and 0.05; 95% CI: 0.01 to 1.00, *p* = 0.05, respectively). By contrast, consumption of dark tea was associated with a decrease in TyG in both unadjusted and multivariable-adjusted models (−0.20; 95% CI: −0.30 to −0.10, *p* < 0.001 and −0.23; 95% CI: −0.34 to −0.13, *p* < 0.001, respectively).

## 4. Discussion

Our prospective study demonstrated that in Chinese community-dwelling adults with prediabetes, daily consumption of dark tea, while not linked to prediabetes regression, was associated with reduced odds of progressing to diabetes over a 3-year period. Conversely, daily consumption of green tea was not associated with prediabetes progression but was linked to a reduced probability of regressing to normoglycemia. In line with these findings, dark tea consumption was associated with a decrease in insulin resistance, whereas green tea was associated with increased insulin resistance. Subgroup analyses suggested potential sex and ethnic differences, with females and certain ethnic groups appearing to be more responsive to habitual tea consumption. These observations are indicative of distinct glycemic impacts of different teas in prediabetes, and support the need for interventional studies to validate the potential of dark tea for prevention of prediabetes progression and to define the implications of high intake of green tea in this context.

To our knowledge, this study is the first prospective evaluation of the associations between tea consumption and prediabetes progression and regression. The significant association between daily dark tea intake and a reduced likelihood of prediabetes progression aligns with cross-sectional evidence linking habitual dark tea consumption to a lower risk of type 2 diabetes [8,9] and, accordingly, provides further support for interventional studies to validate the potential of dark tea consumption to prevent progression of prediabetes. Although our subgroup analyses were constrained by small and uneven sample sizes, the reduced odds of prediabetes progression appeared to be greater in females and non-Han ethnic subgroups, suggesting that the beneficial impact of dark tea is modified by gender and ethnicity. However, dark tea intake was not associated with prediabetes regression, indicating a limited capacity of dark tea to reverse the metabolic impairments in prediabetes.

In contrast to its widely perceived benefits, daily green tea consumption in our study was not associated with prediabetes progression, but was rather associated with reduced odds of prediabetes regression, suggesting a potentially unfavorable effect of green tea in individuals with prediabetes. This finding contrasts with a Japanese cohort study, in which high green tea intake (> or =6 cups per day) was associated with a reduced incidence in type 2 diabetes [16]. Notably, other cohort studies conducted in Chinese [19] and Iranian [20] adults have reported a positive association between higher green tea consumption and increased risk of type 2 diabetes. Similarly, a cross-sectional study in rural Vietnamese adults found that higher green tea intake was related to increased prevalence of type 2 diabetes [18]. The exact reasons behind these discrepancies are not entirely clear, but may reflect, at least in part, population-specific differences. In support of this concept, our subgroup analysis indicated that the reduction in odds of prediabetes regression among green tea drinkers was more evident in females and Han Chinese. These observations underscore the need for further examination into the effects of green tea intake on glucose metabolism, particularly in individuals with prediabetes. In contrast, black tea consumption appeared to have a neutral association with both prediabetes progression and regression, consistent with findings from prior cohort and cross-sectional studies [8,9,35].

The divergent effects of dark, green, and black teas likely reflect differences in their processing methods and resulting bioactive compound profiles [36]. Indeed, various biological actions of different types of tea have been reported. Notably, dark tea is the only variety that undergoes microbial fermentation, and its microbial metabolites are believed to contribute substantially to its unique biological activities [37]. Dark tea and its bioactive compounds, such as flavonoids, theabrownins, phenolics and polysaccharides, have the capacity to enhance antioxidant enzyme activities and hence exert potent antioxidant activities [37,38,39]. In addition, dark tea polysaccharides have been reported to modulate the balance of beneficial and harmful gut microbiota, thereby influencing gut flora function and metabolite production [40]. Theabrownins extracted from Pu-erh tea, a well-known dark tea, have been observed to lower cholesterol and decrease lipogenesis via effects on gut microbiota and bile acid metabolism [41]. Furthermore, dark tea extracts have been reported to improve insulin sensitivity and glucose tolerance by enhancing insulin signaling pathways [39,42] and reducing peripheral insulin resistance [43]. These findings are consistent with our observation that dark tea consumption was associated with lower TyG levels, a surrogate marker for insulin resistance. In a recent study, habitual consumption of dark tea was also found to be associated with an increase in urinary glucose and sodium excretion [9]. In contrast, green tea has been linked to increased insulin resistance [18,44], which aligns with our finding that green tea consumption was associated with elevated TyG levels. These observations may reflect underlying differences in phytochemical composition and metabolic effects between tea types and warrant further mechanistic investigation.

Several limitations should be recognized in interpreting our findings. First, information on tea consumption was obtained by self-reported questionnaire, which inevitably entails a risk of bias. Second, the duration and quantity of tea consumption were not precisely quantified, preventing evaluation of potential dose- and time-dependent effects. Third, potential variations in tea consumption during the 3-year follow-up were not assessed. Fourth, despite adjustments for multiple covariates, residual confounding from unmeasured factors cannot be ruled out. In particular, the consumption of other beverages, such as milk, coffee, sugar-sweetened beverages (SSBs) and artificial sweetener-sweetened beverages (ASBs), which were commonly consumed, may have influenced our results. For example, coffee intake has been linked to a reduced risk of type 2 diabetes [45,46,47], while both SSBs and ASBs are demonstrably associated with increased diabetes risk [48,49,50,51,52]. Fifth, the sample sizes of the subgroups were relatively small, so that type II errors cannot be excluded. Finally, our findings indicate potential sex and ethnic disparities in the glycemic impact of tea consumption. Accordingly, future studies across different populations are warranted to better delineate the benefits and risks associated with specific types of tea.

In conclusion, among Chinese community-dwelling adults with prediabetes, daily consumption of dark tea was associated with a lower likelihood of progressing to type 2 diabetes, while green tea intake was linked to reduced odds of regressing to normoglycemia. These findings support further exploration of dark tea as a strategy to prevent prediabetes progression and the relevance of limiting green tea consumption in this population.

## Figures and Tables

**Figure 1 nutrients-17-02366-f001:**
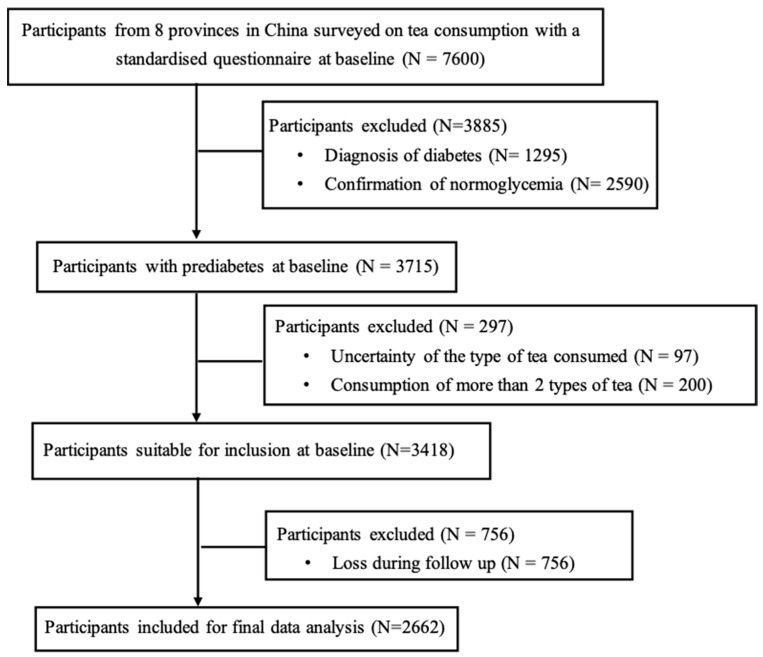
Study flow chart.

**Table 1 nutrients-17-02366-t001:** Differences between green, black and dark tea.

Tea Type	Processing Method	Key Steps	Major Composition Differences
Green tea	Unfermented/unoxidized	Steaming or pan-firing to inactivate enzymes Rolling Drying	High catechin content (especially EGCG) Low theaflavins and thearubigins Retains more polyphenols and vitamin C
Black tea	Fermented/oxidized	Withering Rolling Oxidation Drying	High in theaflavins and thearubigins (from catechin oxidation) Lower catechin levels Darker color and stronger flavor
Dark tea	Post-fermented	Microbial fermentation Aging Drying	High theabrownins and microbial metabolites Very low catechin levels Rich in polysaccharides and unique bioactive compounds

**Table 2 nutrients-17-02366-t002:** Baseline characteristics of participants with prediabetes, who regressed to normoglycemia, remained as prediabetes and progressed to diabetes.

	Regression to Normoglycemia (N = 716)	Remained as Prediabetes (N = 1647)	Progression to Diabetes (N = 299)	*p* _overall_
Gender				<0.001
Male (%)	248 (34.6%)	570 (34.6%)	138 (46.2%)	
Female (%)	468(65.4%)	1077 (65.4%)	161 (53.8%)	
Age (y)	51.9 ± 9.1 ***	54.3 ± 8.2	55.0 ± 8.4	<0.001
BMI (kg/m^2^)	25.1 ± 4.0 **	25.8 ± 4.4	26.1 ± 4.5	<0.001
WC (cm)	84.3 ± 9.4 *	85.5 ± 10.3	81.1 ± 10.8 ***	<0.001
SBP (mmHg)	136.8 ± 19.8 ***	137.1 ± 19.3	144.6 ± 20.4 ***	<0.001
DBP (mmHg)	82.6 ± 11.7	83.2 ± 11.4	86.6 ± 11.4 ***	<0.001
FPG (mmol/L)	5.7 ± 0.4 ***	5.8 ± 0.5	6.1 ± 0.5 ***	<0.001
2hPG (mmol/L)	6.8 ± 1.5 ***	7.3 ± 1.6	8.3 ± 1.9 ***	<0.001
HbA1c (%)	5.3 ± 0.4 ***	5.5 ± 0.4	5.7± 0.4 ***	<0.001
TC (mmo/L)	4.9 ± 1.0 ***	5.1 ± 1.0	5.0 ± 1.0	<0.001
TG (mmo/L)	1.6 ± 1.5 *	1.8 ± 1.9	2.2 ± 2.8 **	<0.001
HDL-C (mmo/L)	1.6 ± 0.4 *	1.5 ± 0.4	1.5 ± 0.4	<0.001
LDL-C (mmo/L)	2.7 ± 0.7 ***	2.8 ± 0.8	2.8 ± 0.8	<0.001
eGFR (mL/min/1.73 m^2^)	99.2 ± 15.1	97.9 ± 13.7	97.7 ± 15.2	0.11
Ethnicity, n (%)				<0.001
Han	595 (83.1%)	1311 (79.6%)	271(90.6%)	
Other	121 (16.9%)	336 (20.4%)	28 (9.4%)	
Vegetable consumption, *n* (%)				0.85
Minimal	3 (0.4%)	5 (0.3%)	2 (0.7%)	
Low	32 (4.5%)	77 (4.7%)	12 (4.0%)	
Moderate	408 (57.0%)	898 (54.5%)	162 (54.2%)	
High	273 (38.1%)	667 (40.5%)	123 (41.1%)	
Fruit consumption, *n* (%)				0.26
Minimal	20 (2.8%)	56 (3.4%)	12 (4.0%)	
Low	329 (45.9%)	721 (43.8%)	133 (44.5%)	
Moderate	339 (47.3%)	774 (47.0%)	145 (48.5%)	
High	28 (3.9%)	95 (5.8%)	9 (3.0%)	
Low-salt and low-fat diet, *n* (%)	224 (31.3%)	491 (29.8%)	94 (31.4%)	0.71
Regular exercise, *n* (%)	203 (28.4%) ***	512 (31.1%)	76 (25.4%)	0.09
Hypotensive medication, *n* (%)	135 (18.9%) ***	437 (26.5%)	119 (39.8%) ***	<0.001
Family history of diabetes, *n* (%)	110 (15.4%)	289 (17.5%)	59 (19.7%)	0.20
Current smoker, *n* (%)	116 (16.2%)	273 (16.6%)	77 (25.8%) ***	<0.001
Habitual alcohol drinker, *n* (%)	148 (20.7%) **	365 (22.2%)	79 (26.4%)	0.13

Abbreviations: BMI: body mass index; WC: waist circumference; SBP: systolic blood pressure; DBP: diastolic blood pressure; FPG: fasting plasma glucose; HbA1c: hemoglobin A1c; TC: total cholesterol; TG: triglyceride; LDL-C: low-density lipoprotein cholesterol; HDL-C: high-density lipoprotein cholesterol; eGFR: estimated glomerular filtration rate. Data are presented as means ± standard deviations or numbers (%), where appropriate. One-way analysis of variance was used for comparisons of continuous variables, while the χ^2^ test was used for comparison of categorical variables. Post hoc comparisons were adjusted using Bonferroni’s correction. * *p* < 0.05 and ** *p* < 0.01, *** *p* < 0.001, compared to participants with prediabetes.

**Table 3 nutrients-17-02366-t003:** Tea consumption and odds of prediabetes progression and regression.

	N, Cases/Total	Model 1		Model 2	
		OR (95% CI)	*p*	OR (95% CI)	*p*
Progression to Diabetes					
Type of Tea					
No tea	126/1198	1.00 (Ref.)		1.00 (Ref.)	
Green tea	141/1016	1.27 (0.97, 1.65)	0.08	1.10 (0.83, 1.47)	0.50
Black tea	9/138	0.59 (0.29, 1.20)	0.15	0.61 (0.29, 1.27)	0.19
Dark tea	5/140	0.30 (0.12, 0.75)	0.01	0.28 (0.11, 0.72)	0.01
Other	18/170	1.13 (0.66, 1.94)	0.66	1.03 (0.59, 1.80)	0.92
Regression to Normoglycemia					
Type of Tea					
No tea	344/1198	1.00 (Ref.)		1.00 (Ref.)	
Green tea	232/1016	0.76 (0.63, 0.93)	0.01	0.73 (0.59, 0.90)	0.01
Black tea	41/138	0.99 (0.67, 1.46)	0.94	0.98 (0.65, 1.48)	0.92
Dark tea	39/140	0.86(0.58, 1.27)	0.45	0.81 (0.53, 1.24)	0.34
Other	60/170	1.38 (0.97, 1.96)	0.07	1.19 (0.82, 1.71)	0.36

Multinominal logistic regression analysis was employed to assess the associations of tea consumption with prediabetes progression and prediabetes regression. Model 1: The crude model. Model 2: Adjusted for age, sex, body mass index, mean arterial pressure, triglyceride, high-density lipoprotein-cholesterol, low-density lipoprotein-cholesterol, estimated glomerular filtration rate, low-salt low-fat diet, regular exercise, hypotensive medication, family history of diabetes, current smoking status, current alcohol consumption, vegetable consumption, and fruit consumption.

**Table 4 nutrients-17-02366-t004:** Frequencies of different tea consumption and prediabetes progression and regression.

	N, Cases/Total	Model 1		Model 2	
		OR (95% CI)	*p*	OR (95% CI)	*p*
Progression to Diabetes					
Frequency of Green Tea Consumption					
Never	158/1646	1.00		1.00	
Sometimes	58/473	1.21 (0.87, 1.68)	0.25	1.17 (0.83, 1.63)	0.37
Daily	83/543	1.56 (1.16, 2.09)	0.003	1.27 (0.92, 1.76)	0.14
Frequency of Dark Tea Consumption					
Never	294/2522	1.00		1.00	
Sometimes	2/31	0.42 (0.10, 1.79)	0.24	0.49 (0.11, 2.15)	0.35
Daily	3/109	0.22 (0.07, 0.71)	0.01	0.22 (0.07, 0.71)	0.01
Regression to Normoglycemia					
Frequency of Green Tea Consumption					
Never	484/1646	1.00		1.00	
Sometimes	111/473	0.76 (0.59, 0.97)	0.03	0.74 (0.58, 0.95)	0.02
Daily	121/543	0.74 (0.59, 0.94)	0.01	0.72 (0.56, 0.92)	0.01
Frequency of Dark Tea Consumption					
Never	677/2522	1.00		1.00	
Sometimes	4/31	0.37 (0.13, 1.06)	0.06	0.31 (0.10, 0.91)	0.03
Daily	35/109	1.13 (0.75, 1.71)	0.57	1.14 (0.73, 1.78)	0.58

Multinominal logistic regression analysis was employed to assess the associations of tea consumption with prediabetes progression and prediabetes regression. Model 1: The crude model. Model 2: Adjusted for age, sex, body mass index, mean arterial pressure, triglyceride, high-density lipoprotein-cholesterol, low-density lipoprotein-cholesterol, estimated glomerular filtration rate, low-salt low-fat diet, regular exercise, Hypotensive medication, family history of diabetes, current smoking status, current alcohol consumption, vegetable consumption, and fruit consumption.

**Table 5 nutrients-17-02366-t005:** Association of tea consumption insulin resistance.

	TyG			
	Model 1		Model 2	
	Mean (95% CI)	*p*	Mean (95% CI)	*p*
Type of Tea				
No tea	0.00 (Ref.)		0.00 (Ref.)	
Green tea	0.08 (0.03, 0.13)	0.001	0.05 (0.01, 1.00)	0.05
Black tea	−0.04 (−0.14, 0.06)	0.43	−0.06 (−0.17, 0.04)	0.21
Dark tea	−0.20 (−0.30, −0.10)	<0.001	−0.23 (−0.34, −0.13)	<0.001
Other	0.08 (−0.14, 0.17)	0.10	0.05 (−0.04, 0.15)	0.25

Linear regression analysis was employed to assess the associations of tea consumption with insulin resistance. Model 1: The crude model. Model 2: Adjusted for age, sex, body mass index, mean arterial pressure, estimated glomerular filtration rate, low salt low-fat diet, regular exercise, antihypertensive medication, family history of diabetes, current smoking status, current alcohol consumption, vegetable consumption, and fruit consumption.

## Data Availability

The original contributions presented in the study are included in the article/Appendix A, further inquiries can be directed to the corresponding authors.

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
