# Peer review of "The Impact of Tea Consumption on Prediabetes Regression and Progression: A Prospective Cohort Study"

_nutrients, 2025, doi:10.3390/nu17142366_

Round 1

Reviewer 1 Report

Comments and Suggestions for Authors

This study investigated the associations of tea consumption with prediabetes progression and regression in Chinese adults with prediabetes. A cohort of 2662 Chinese adults with prediabetes was followed over 3 years. Baseline tea consumption, including the type (green, black, dark and other) and frequency (sometimes, daily, or nil) was assessed using standardized questionnaires. Compared to non-tea drinkers, dark tea consumers were less likely to progress to T2DM, whereas green tea consumers was associated with a reduced probability of regression to normoglycemia.

The study is clinically of interest. The prospective study design is a strength of the study. The findings are shown clearly and the conclusion is supported by the data. There are some comments.

  1. Although the authors mentioned as a limitation, lack of information regarding other drinks including coffee and sugar-contained beverages is critical.

  1. The uneven sample size among the groups is another limitation. Baseline patient characteristics according to types of tea (green, dark and black) should also be presented in a table.

Reviewer 2 Report

Comments and Suggestions for Authors

This study focused on the topic of tea consumption impact on prediabetes condition. However, I must express my concerns regarding this study.

 The introduction is very short, please mention other studies focused on the role of tea consumption on metabolic balance. Please elaborate on the topic of tea composition and its relationship with diabetes.

Was there available any information about other food/drink products along the study? It should be included in the analysis as there can be some other associated aspects related. To observed trends.

Line 79 please provide details of the standardized questionnaire used.

There was also presented impact of BMI and other parameters on diabetic condition, please provide all details of the models that acknowledged them in multivariable models?

If some other data regarding diet is available, please provide additional analyses.

The discussion is very short. The number of references is low, please elaborate on the topic and refer to more other studies.

Reviewer 3 Report

Comments and Suggestions for Authors

This manuscript presents a well-conducted prospective cohort study exploring the role of tea consumption particularly dark and green tea on the progression and regression of prediabetes in a Chinese adult population.

While the statistical associations are compelling, the biological plausibility of why dark tea may reduce progression to type 2 diabetes while green tea appears to reduce the chance of regression to normoglycemia requires further explanation. A brief discussion of potential phytochemical or gut microbiota-related mechanisms would strengthen the conclusion.

The finding that green tea may reduce the chance of regression to normoglycemia is counterintuitive based on previous literature. This warrants more critical discussion and exploration of whether this is a spurious finding, a population-specific effect, or a reflection of unknown variables.

Accept with Minor Revisions: subject to clarification of mechanisms, limitations, and interpretation of the green tea findings.

Round 2

Reviewer 1 Report

Comments and Suggestions for Authors

The manuscript has been revised properly.

Author Response

The manuscript has been revised properly.

We sincerely appreciate your positive evaluation of our revised manuscript. We are pleased that the revisions have addressed your concerns.

Reviewer 2 Report

Comments and Suggestions for Authors

Despite their efforts, the Authors failed to sufficiently improve the quality of the manuscript to suit the requirements of the journal.

The Authors indicate that important confounders that could impact the results were not assessed within the study which compromises the validity of the findings.

Line 156 – how was it assessed – in portions, grams, quartiles?

Table 3 – regarding tea intake measures – “sometimes” is not defined and is not accurate, moreover among daily intake it can also vary. The main exposure factor was also not enough carefully measured.

Author Response

Reviewer 2

Despite their efforts, the Authors failed to sufficiently improve the quality of the manuscript to suit the requirements of the journal.

We appreciate your continued engagement and the opportunity to further improve our manuscript based on your constructive comments. Below, we respond to each specific point.

  1. The Authors indicate that important confounders that could impact the results were not assessed within the study which compromises the validity of the findings.

We acknowledge this important concern. As stated in the manuscript’s limitations section, not all potential confounding variables could be captured due to the constraints of our dataset. This is a common limitation of observational research. However, we made every effort to include and adjust for all available and relevant confounders in our statistical models. These adjustments were aimed at enhancing the internal validity of our findings, which we have now clarified more explicitly in the revised Discussion section (Lines 302-317).

  1. Line 156 – how was it assessed – in portions, grams, quartiles?

Thank you for highlighting this. We have now clarified that fruit and vegetable intake was assessed using a structured questionnaire, with participants categorizing their consumption as “Minimal”, “Low”, “Moderate”, or “High” (Lines 104-105). While we acknowledge that this categorical approach lacks the precision of portion-based quantification (e.g., grams or servings), it reflects the retrospective, self-reported nature of the data and helps mitigate recall bias. Additional details are provided in Table 1 and Supplementary Table 1. This variable has been included as a covariate in our adjusted models to reduce potential confounding.

  1. Table 3 – regarding tea intake measures – “sometimes” is not defined and is not accurate, moreover among daily intake it can also vary. The main exposure factor was also not enough carefully measured.

We appreciate this point. To improve clarity, we have explicitly defined the term “sometimes” in the Methods section (Lines 126–128), specifying its frequency range based on participant responses. We agree that variability in individual tea consumption, especially among daily drinkers, may not be fully captured. As such, we have addressed this as a study limitation (Lines 304–305). While our primary exposure variable was based on tea type and frequency, which may not capture quantity or timing, we believe this classification still offers meaningful insights into general consumption patterns. As shown in Table 3, our analysis revealed distinct associations for dark and green tea with prediabetes outcomes. These findings, although preliminary, offer valuable leads for future, more detailed investigations, as emphasized in Lines 315-317.